



# Predictability of the variable solar-terrestrial coupling

Ioannis A. Daglis[1,15], Loren C. Chang[2], Sergio Dasso[3], Nat Gopalswamy[4], Olga V. Khabarova[5], Emilia Kilpua[6], Ramon Lopez[7], Daniel Marsh[8,16], Katja Matthes[9,17], Dibyendu Nandi[10], Annika Seppälä[11], Kazuo Shiokawa[12], Rémi Thiéblemont[13] and Qiugang Zong[14]

[1]Department of Physics, National and Kapodistrian University of Athens, 15784 Athens, Greece
[2]Department of Space Science and Engineering, Center for Astronautical Physics and Engineering, National Central University, Taiwan
[3]Department of Physics, Universidad de Buenos Aires, Buenos Aires, Argentina
[4]Heliophysics Science Division, NASA Goddard Space Flight Center, Greenbelt, MD 20771, USA
[5]Solar-Terrestrial Department, Pushkov Institute of Terrestrial Magnetism, Ionosphere and Radio Wave Propagation of RAS (IZMIRAN), Moscow, 108840, Russia
[6]Department of Physics, University of Helsinki, Helsinki, Finland
[7]Department of Physics, University of Texas at Arlington, Arlington, TX 76019, USA
[8]National Center for Atmospheric Research, Boulder, CO 80305, USA
[9]GEOMAR Helmholtz Centre for Ocean Research, Kiel, Germany
[10]IISER, Kolkata, India
[11]Department of Physics, University of Otago, Dunedin, New Zealand
[12]Institute for Space-Earth Environmental Research, Nagoya University, Nagoya, Japan
[13]LATMOS, Universite Pierre et Marie Curie, Paris, France
[14]School of Earth and Space Sciences, Peking University, Beijing, China
[15]Hellenic Space Center, Athens, Greece
[16]Faculty of Engineering and Physical Sciences, University of Leeds, Leeds, UK
[17]Christian-Albrechts Universität, Kiel, Germany

*Correspondence to*: Ioannis A. Daglis (iadaglis@uoa.gr)

**Abstract.** In October 2017, the Scientific Committee on Solar-Terrestrial Physics (SCOSTEP) Bureau established a
committee for the design of SCOSTEP's Next Scientific Program (NSP). The NSP committee members and authors of this paper, decided from the very beginning of their deliberations that the predictability of the Sun-Earth System from a few hours to centuries is a timely scientific topic, combining the interests of different topical communities in a relevant way. Accordingly, the NSP was christened PRESTO – **Pre**dictability of the variable **S**olar-**T**errestrial c**o**upling. This paper presents a detailed account of PRESTO; we show the key milestones of the PRESTO roadmap for the next five years, review
the current state of the art and discuss future studies required for the most effective development of solar-terrestrial physics.

## 1 Introduction

The Sun is a variable star and its variability influences the Earth's space and atmospheric environment in a variety of ways, some of them being rather drastic and leading to profound changes in the properties of fields and particles in geospace (e.g.,
Daglis et al., 2019). Varying solar and interplanetary magnetic fields and solar wind plasma parameters, radiative flux and energetic particle enhancements force the terrestrial magnetosphere, ionosphere, atmosphere and climate, leading to dramatic effects. Transient energetic events such as flares, coronal mass ejections (CMEs), interplanetary shocks (ISs), stream and corotating interaction regions (SIRs/CIRs), and energetic particles - both from the Sun and within the Earth's radiation belts - adversely impact critical technologies based in space and on Earth that our society is increasingly dependent upon (e.g.,
Bothmer and Daglis, 2007; Eastwood et al., 2018). At the same time, the middle and upper atmosphere/ionosphere are impacted by processes originating at lower altitudes, e.g., by atmospheric gravity waves, tides and planetary waves, and changes in radiatively active gases (Qian and Solomon, 2012; Oberheide et al., 2015; Baldwin et al., 2018). With the





understanding of causal connections in the Sun-Earth system maturing over the last several decades, fuelled by both observations and theoretical modelling, we are in a position to begin the transition of this knowledge to predictions of the

Sun-Earth coupled system of relevance to society. PRESTO (**Pre**dictability of the variable **S**olar-**T**errestrial c**o**upling), the new scientific programme of SCOSTEP, aims at facilitating this interdisciplinary endeavour through focused, internationally coordinated efforts addressing predictability of the Sun-Earth system variabilities ranging across space weather and climate timescales (as illustrated in Figure 1). Synergies with existing national and international research programs are also encouraged.

PRESTO addresses the predictability of:

1. Space weather on timescales from seconds to days and months, including processes at the Sun, in the heliosphere and in the Earth's magnetosphere, ionosphere, and atmosphere,

2. Sub-seasonal to decadal and centennial variability of the Sun-Earth system, with a special focus on climate impacts and a link to the World Climate Research Program Grand Challenge Near-Term Climate Predictions as well as the

Intergovernmental Panel on Climate Change.

A major motivation for PRESTO is the desire to conduct fundamental research that has the promise to advance predictive capability with societal implications. Extreme events, such as Carrington event size solar eruptions and geospace storms, have attracted particular attention lately (e.g., Baker et al., 2013; Gopalswamy, 2018; Knipp et al., 2018; Hayakawa et al., 2019). They are rare (occurring once in a century or even less frequently) but if they occur and impact the Earth, they

can have potentially devastating effects on the modern technology infrastructure in space and on ground. Strong and intense storms occur a few times per solar cycle and can likewise have significant, although less deleterious consequences (e.g., spacecraft anomalies, the loss of transformers). In addition, moderate storms occurring on a monthly basis also matter and their space weather effects are important to understand, predict and mitigate, in particular long-term exposures (e.g., corrosion on pipelines, spacecraft charging). An important thing to also keep in mind is that different manifestations of

space weather in different domains do not necessary occur always simultaneously or with same magnitude. On longer timescales, there is a pressing need to be able to separate natural from anthropogenic forcing of Earth's climate (Emmert et al., 2012; Yue et al., 2015) and extend the length of useful weather forecasts (National Academies of Sciences, 2016).

Predicting the Sun-Earth system variabilities as a whole is highly challenging. Besides the different timescales discussed previously, this topic covers non-linear and multi–scale phenomena in highly different plasma and neutral fluid

domains that are often coupled in a complex way from the Sun to the Earth's atmosphere and oceans. Furthermore, different communities in the field are often separate, and use different models, terminology and approaches in their studies. It is hoped that by selecting predictability as an overarching theme for PRESTO, it will encourage the scientific community to view the various sub-domains within solar-terrestrial physics as part of a chain within a coupled system, as illustrated in Figure 1. By better understanding this chain and its various links, we aim to improve prediction of phenomena that have significant

societal relevance. Advancement in this area will require improved synthesis of observations and models, along with improvements in tools such as data assimilation and statistical analysis. It is hoped that viewing the problem in terms of timescales will foster a more interdisciplinary view and increase international collaboration.

PRESTO is aligned along three pillars of research:

Pillar 1. Sun, interplanetary space and geospace

Pillar 2. Space weather and the Earth's atmosphere

Pillar 3. Solar activity and its influence on the climate of the Earth System

In the following sections, specific areas of scientific focus, grouped by pillar, are listed, where progress needs to be achieved to significantly improve our predictive skill of the solar-terrestrial system.



## 2 Sun, interplanetary space and geospace

The properties of geoeffective solar and heliospheric events, such as coronal mass ejections (CMEs), interplanetary coronal mass ejections (ICMEs), interplanetary shocks (ISs), stream and corotating interaction regions (SIRs/CIRs), SEPs (Solar Energetic Particles), the consequent solar wind – magnetosphere coupling and the internal magnetospheric dynamics play complex and intertwined roles in geospace weather. Accurate and reliable predictions of geospace weather (including the dynamics of the various kinds of energetic particles and of plasma waves in the inner magnetosphere) require the understanding of the key aspects of the complex interplay of external and internal regulating factors operating over timescales ranging from minutes to days. Most space weather end-users need long-lead time predictions, i.e., warnings given at least half a day in advance (and preferably longer). With the current instrumentation this means estimating the impact based on remote-sensing observations and modelling using them, to give estimate occurrence and properties of CMEs/ICMEs, CIRs/SIRs and their embedded magnetic structures, interplanetary shocks, fast streams and properties of the background solar wind that CMEs propagate within. However, to achieve this a lot of uncertainties in the observed parameters that are used to feed propagation and acceleration models will need to be addressed.

### 2.1 Solar and Interplanetary Drivers

### 2.1.1 Predictability of Coronal Mass Ejections and solar flares

Predicting the occurrence of solar flares and CMEs, and the arrival times and properties of Earth-impacting ICMEs, are major challenges. The relevant timescales vary from seconds to hours, for flares and the eruption of CMEs, to the several days it takes for a CME to propagate from the Sun to the Earth. The frequency and properties of CMEs and flares also vary in accordance with the Sun's 11-year activity cycle and overall solar activity levels (e.g., Lamy et al., 2019; Gopalswamy et al. 2020). While the occurrence rate of flares and CMEs of moderate to strong size increases with increasing solar cycle strength, it is still an open question how the occurrence of the most extreme eruptions correlates with the solar cycle amplitude (e.g., Kilpua et al., 2015b). Investigation of the solar flare spectral irradiance is also required to define better the input for ionospheric variability models and societal impact.

The formation of the eruptive structures at the Sun can take from hours to days, but their destabilization is a fast process, occurring when the energy stored in highly sheared or twisted field along magnetic photospheric polarity inversion lines (PILs) is rapidly released by some form of magnetic reconnection (e.g., Green et al., 2018; Welsch, 2018). In some cases, the eruptions result in geoeffective SEP events when the associated CMEs drive fast mode MHD shocks. The occurrence probability of flares and CMEs depends on properties of sunspots and solar active regions, such as the degree of non-potentiality of the magnetic field and the amount of magnetic helicity (i.e., how twisted, linked and sheared the magnetic field lines are). It is the current consensus that magnetic flux ropes are an integral ingredient of erupting CMEs. However, there is no model currently that predicts when a flare and/or a CME will occur, and we do not yet understand adequately the mechanisms that trigger and drive the eruption. This is particularly problematic from the space weather point of view, because X-ray/EUV emission and highly energetic protons arrive at Earth on a timescale of only minutes following flare onset. onset. A considerable work forward has however made in recent years in this topic using empirical predictors (e.g., Kontogiannis et al., 2018), machine learning (Florios et al., 2018) and physics-based approaches (Kusano et al., 2020).

The time needed for an ICME to arrive at Earth varies from about half-a-day to a few days after the eruption at the Sun, depending on the initial speed of the CME and the speed of the ambient solar wind (Luhmann et al., 2020). Continuous in-situ observations of geoeffective CMEs and CIRs/SIRs are available from the spacecraft located at the gravitational



equilibrium point at the distance of ~0.01 AU from the Earth to the Sun (the so-called L1 Lagrangian point). While the corresponding data are available online within several minutes[1], it takes solar wind streams and CMEs only ~30-80 minutes to reach the terrestrial bow shock from there. As a result, only short-term forecasts with an advance time from tens of minutes to one-three hours have a rather good accuracy (e.g., see Ji et al., 2012 and some relevant websites[2,3,4]). Unfortunately, the accuracy of mid-term and long-lead-time space weather predictions still remains very modest[5]. Regarding

CIR/SIRs more attention needs to be paid to models predicting the geoeffectiveness of CIR/SIRs and solar wind flows (e.g., Rotter et al., 2012). For CMEs the key challenges in making longer term predictions are the following  (e.g., Kilpua et al., 2019):

1. Accurate estimation of the initial properties of CMEs (acceleration, speed, geometrical parameters, propagation direction, flux rope magnetic properties) from on-disk and off-limb observations, and from factors such as the coronal field

structure in the neighbourhood of the eruption.

2. Accurate estimation of when and how CME properties will evolve during their propagation from the Sun to the Earth, and when they will impact the Earth. This includes consideration of how CMEs can be altered during propagation through interactions with the ambient solar wind, e.g., high speed solar wind streams from coronal holes and with other CMEs[6] (Gopalswamy et al. 2000; Wang et al. 2014); such interactions can significantly deform, erode, deflect and rotate

CMEs (e.g., Manchester et al., 2017).

3. Prediction of the properties of the sheath regions of CMEs.

CME and CIR/SIR's kinematic and geometrical parameters and their propagation direction can be estimated from remote-sensing tomography or heliospheric imagery and related reconstruction techniques. Reconstructions of key plasma and interplanetary magnetic field (IMF) parameters of the stream as it flows away from the solar corona to ~3 AU are

available from heliospheric imagers on board of spacecraft that observe the solar wind in white light[7,8] (Jackson et al., 2009). An alternative way is to employ ground-based observations of interplanetary scintillations that represent fluctuations in the intensity of radio sources caused by solar wind structures propagating through the line of sight[9] (Bisi et al., 2010). Since scintillations of numerous radio sources are observed simultaneously, this allows reconstructing 3-D maps of the dynamical solar wind. Although the measurements are often subject to significant projection effects, and the resolution of reconstructed

images is rather low (not better than 0.05 AU in height and 1°x1° in latitude and longitude), the corresponding 3-D reconstructions are very helpful in understanding the global situation in the interplanetary medium full of simultaneously existing steams/flows of various origin that propagate with varying velocities. These techniques also allow studying ICME-SIR/CIR interactions that lead to changing the form and trajectories of the both interacting objects (Khabarova et al. 2016; Malandraki et al. 2019).

It is accepted that the direction and flux rope magnetic properties of the resulting CME are strongly shaped by coronal magnetic structures of the erupting system and in its immediate vicinity (Patsourakos and Georgoulis 2017; Gopalswamy et al., 2018a).  However, due to the lack of observations of the magnetic field within the CME and in the surrounding corona, determining these propagating CME properties is particularly challenging for forecasting (the so called "$B_z$ challenge"). Current attempts to estimate the magnetic field in CME flux ropes include using indirect solar proxies

(based on EUV and X-ray observations and magnetograms) and data-driven coronal modelling (Gopalswamy et al., 2018b;

[1] https://www.swpc.noaa.gov/products/real-time-solar-wind
[2] https://www.swpc.noaa.gov/sites/default/files/images/u30/G2%20%28K6%29%20Warnings.pdf
[3] https://www.swpc.noaa.gov/sites/default/files/images/u30/G2%20%28K6%29%20Warnings.pdf
[4] http://spaceweather.ru/forecast
[5] https://www.swpc.noaa.gov/sites/default/files/images/u30/Max%20Kp%20and%20GPRA.pdf
[6] https://swrc.gsfc.nasa.gov/main/cmemodels
[7] http://smei.ucsd.edu/new_smei/index.html
[8] http://helioweather.net/archive/
[9] https://ips.ucsd.edu/



Sarkar et al., 2020). The magnetic field direction and strength dictate how effectively magnetic reconnection between the interplanetary and geomagnetic field develops. Because the timing of a geomagnetic storm depends on what part of the ICME holds the strongest southward magnetic field, the above-noted uncertainties in magnetic properties currently results in differences of up to one day in estimates of storm-occurrence times.

In recent years, there has been substantial improvement in predicting CME Earth-arrival times using numerical first-principle simulations, e.g., ENLIL (Odstrcil, 1999), EUHFORIA (Pomoell and Poedts, 2018), SUSANOO (Shiota and Kataoka, 2016). There has also been progress in semi-empirical/analytical models, e.g., Drag Based Model (DBM; Vršnak et al., 2013, combined with observational techniques, such as interplanetary scintillation (e.g., Park et a., 2020), wide-angle heliospheric imaging (e.g., Möstl et al., 2017), and radio waves generated at the CME shocks (e.g., Cane et al., 1982;

Magdalenić et al., 2014), as well as some adaptive numerical methods. Much of these advances have benefited from dedicated "campaign studies", as well as real-time-prediction services, both of which rely upon close interaction and communication between modelling and observations. However, there is currently no model that consistently makes accurate predictions of the ICME arrival times and impact details (i.e., whether a CME will make a direct hit or a glancing blow with Earth) and captures details of all major CME deformations. For example, the drag force of the ambient solar wind on CMEs

can vary substantially from case to case, and numerical simulations are not yet routinely run with flux rope CMEs, thus lacking capability to predict accurately their magnetic properties and interactions. As a consequence of the evolution and interactions, ICMEs may have a highly complex structure (e.g., Manchester et al., 2017). For example, a flux rope may not be present at all, or it might occupy only a part of a distorted ICME structure, for instance as a consequence of erosion and magnetic flux removal (Dasso et al., 2006). Multiple CMEs can also merge to form "complex ejecta" where the

characteristics of the individual CMEs are lost, or the following CME can strongly compress the field of the leading CME (e.g., Burlaga, Plunkett and St. Cyr. 2002; Lugaz et al., 2017); the latter case can result in particularly severe space weather effects (e.g., Liu et al., 2014). Another recently highlighted question for space weather is how coherent CME flux ropes are. Several studies suggest that the properties of a CME flux rope (e.g., their orientation) may change considerably over relatively small longitudinal separations (about a few degrees) (e.g., Owens, Lockwood and Barnard, 2017; Lugaz et al.,

2018; Good et al., 2019). Recently flux rope and spheromak models have been implemented in numerical simulations which are expected to improve significantly in their capability to predict geospace disturbances. One such case is presented in Figure 2 that shows the snapshot of the EUHFORIA run with the spheromak model of three interacting CMEs (Scolini et al., 2020). Interactions were found to significantly affect storm intensity and arrival time.

Regarding ICME sheaths, i.e., the turbulent wakes generated by the ICME shock downstream, the critical issue is

their high turbulence and the strong internal variations (e.g., Kilpua, Koskinen and Pulkkinen, 2017). Sheaths can drive major geospace storms independent of whether the CME flux rope will be geoeffective, and they have particularly strong effects at high latitudes (e.g., Huttunen et al, 2004; Guo et al., 2011, and references therein). In fact, one of the extreme geomagnetic storms (Halloween 2003 period) was entirely due to the sheath, while the flux rope was not geoeffective (Gopalswamy, 2008). Currently, there is no practical way to estimate sheath properties in advance, although there are

theoretical works that analyze a thickness of interplanetary sheaths, an arrival time of the structure to the Earth and its possible impact on the magnetosphere (e.g., Takahashi and Shibata 2017). Furthermore, for accurate understanding of sheaths we have to determine whether magnetohydrodynamic (MHD) models are capable of predicting their turbulent properties, or if a kinetic/hybrid approach is required. Resolving in more detail the internal sheath structure, and determining how it depends on the driver (CME and shock) and the ambient solar wind may also help in predicting their space weather

response.

Predictions of geoeffectiveness of ICMEs and ICME sheaths also suggest an analysis of the role of ICME-driven interplanetary shocks (ISs). Details are given in Section 2.1.3 below. Here, we just stress the importance of a combined consideration of ICME sheaths together with ISs since any sheath, including a planetary magnetosheath, is a shock-borne





structure, and its properties are determined by features of propagation of the corresponding shock (see details in Siscoe and
Odstrcil, 2008). A sheath downstream of the IS is often treated as its turbulent wake filled with waves, discontinuities,
current sheets and 3-D flux ropes/blobs/plasmoids or their 2-D counterparts, magnetic islands (Khabarova et al., 2016; Ala-
Lahti et al., 2018). As a result, dynamical processes and stochastic magnetic reconnection occur in this turbulent region,
leading to specific effects such as the intensification of local particle acceleration due to a combination of different
mechanisms that energize particles (e.g., Zank et al. 2015). The latter effect often overlaps with large SEP events
(Khabarova and Zank, 2017; Malandraki et al. 2019), which overall represents a serious threat to spacecraft/satellite
equipment and may have negative biological and technological consequences (see Section 2.1.3).

Finally, a fraction of space-weather-relevant CMEs are not well observed, due to a lack of typical low-coronal pre-
eruption and eruption-time signatures. These so-called "stealth CMEs" sometimes lead to "problematic geomagnetic
storms", which are storms for which the cause (in this case, the source stealth CME) is not obvious (e.g., Nitta and Mulligan,
2018). Observational, theoretical, and numerical simulation studies are essential to understanding the mechanism(s)
triggering stealth CMEs, and their propagation characteristics in the ambient solar wind

### 2.1.2 Predictability of CIRs/SIRs and their interaction with ICMEs

In section 2.1.1 we showed the importance of studying properties and dynamics of CMEs/ICMEs and the ICME sheath.
Meanwhile, very similar turbulent and compressed sheath-type regions are formed in the solar wind as a result of the
interaction of high-speed flows from coronal holes (CHs) with the ambient slower solar wind. As mentioned above, these
rotating dense regions are called SIRs/CIRs (e.g., Richardson, 2018). CIR's life-time may exceed one solar rotation, and
SIRs are shorter-living analogues of CIRs. The both types of structures are powerful objects in the solar wind, carrying
electric currents and energy in the inner heliosphere, because of their long-lived solar sources and formation of ISs and
strong current sheets at their borders.

Note that ISs similar by properties to ICME-driven shocks are formed at edges of SIRs/CIRs not in the corona, as in
the CME case, but at 2-3 AU. At the Earth's orbit, SIR/CIR-driven shocks are not strong and even not completely formed in
many events, therefore their direct impact on the terrestrial magnetosphere is lesser than the impact of ICME-driven shocks
that propagate outward the Sun and hit the Earth more often. However, with distance, SIRs/CIRs become more powerful
than weakening ICMEs. Because of the poorly formed forward shock at 1 AU, SIRs/CIRs are responsible for the
development of geomagnetic storms without sudden commencements and are often not considered seriously although they
can bring a strong negative vertical component of the IMF to the Earth and consequently cause as strong geomagnetic storms
as ICMEs (e.g. Chi et al. 2018).

Since CHs can appear at any time throughout the solar cycle, and because their shape is irregular and evolves with
time, forecasting CIR/SIR encounters with the Earth's magnetosphere represents a particularly formidable task. Such a
forecast however is critically important, as CIRs/SIRs and fast solar wind streams are the main triggers of geomagnetic
storms in the absence of solar active regions and CMEs. As noted above, CIR/SIR-driven ISs represent a major source of
energetic particles in the heliosphere during solar minima, and they are highly important for causing acceleration of electrons
to relativistic energies in the Van Allen radiation belts. These are all crucial space weather considerations that cannot be
ignored. Therefore, there are additional challenges in studying and predicting space weather effects caused by the impact of
geoeffective CIRs/SIRs and subsequent fast flows from CHs on the terrestrial magnetosphere.

CHs often co-exist with low-latitude active regions, especially around solar maximum, resulting in complex and
poorly-investigated effects as streams and flows on their way to the Earth. Observations in white light show that ICMEs are
often compressed and deflected by CIRs/SIRs, and vice versa, a free flowing SIR can be interrupted by an ICME with the
consequent formation of a compressed ICME-SIR conglomerate at one of ICME flanks (see Figure 3). One may suggest that
these interactions can significantly enhance the geoeffectiveness of both structures, leading to particularly strong and





complex geospace responses. We illustrate this fact below. The three panels in Figure 3a show co-existing ICMEs and SIRs in the interplanetary space as predicted by ENLIL real-time simulations (see the corresponding animation at https://www.wired.com/images_blogs/wiredscience/2012/03/coronal-mass-ejection-forecast-march-5-8-2012-nasa.gif and

details of the technique at https://ccmc.gsfc.nasa.gov/models/modelinfo.php?model=ENLIL; Odstrcil, 2003). This picture is typical for the solar maximum period when coronal holes are located at low heliolatitudes.

The SIRs in Figure 3a resemble elongated sleeves rotating anti-clockwise, and ICMEs can be identified as half-circle-shaped structures propagating more or less radially. The sequence of snapshots shown from top to bottom of Figure 3a allows us to identify two ICMEs ejected with some delay in the background of four long-lived SIRs. Since the second ICME

with a smaller leading front propagates faster, the ICMEs merge beyond the Earth orbit, covering at least a half of the heliosphere. The left panels depict the evolution of the streams and flows in the ecliptic plane (the view is from the North pole of the Sun) and the right panels indicate the shape and dynamics of the streams and flows in the vertical cut made through the Earth position (a yellow dot). According to the ENLIL modelling, the Earth happens to be on the way of the flank of the first ICME actively interacting with a SIR, therefore it meets one of the SIRs first and then an intense density

disturbance of the merged ICME-SIR leading fronts are seen. The ICME is distinguishable as a black contour in the colour SIR background.

The simulations in Figure 3a show that both the high-speed streams and flows from coronal holes push the heliospheric current sheet (HCS) in front of them, which creates one of the most often observable types of magnetic cavities in the heliosphere, namely, the cavities formed by the strongest current sheet at the leading edge of a stream/flow and the HCS

(Khabarova et al. 2015, 2016). The fact that the SIR and ICME leading fronts can act as magnetic walls was debated just ten years ago but is commonly accepted nowadays (see Khabarova et al., 2016; Malandraki et al., 2019; Adhikari et al., 2019; Cécere et al., 2019 and references therein). Recently, it has been found that magnetic cavities play a critical role in the confinement of energetic particles in the solar wind that leads to development of a cascade of effects that allow significant particle acceleration in the heliosphere (see Section 2.1.3).

It is obvious from Figure 3a that SIRs do not allow ICMEs to propagate freely, deflecting those and changing their trajectory, and, on the other hand, the model predicts that the SIR front can be broken by an ICME. One can find a lot of similar movies from ENLIL reconstructions of the solar wind density and the other plasma parameters; see http://www.helioweather.net/, https://ccmc.gsfc.nasa.gov/RoR_WWW/enlil-rt/latest/density.html, and https://iswa.gsfc.nasa.gov/IswaSystemWebApp/ (choose Heliosphere - ENLIL density, velocity and dynamic pressure in the

inner heliosphere)).

Direct observations of the solar wind in white light available from heliospheric imagers (HIs) suggest even more complex combination of events occurring in the inner heliosphere during the same period. Figure 3b shows reconstructions of the density from HI onboard of STEREO B as presented in the Heliowheather archive[10]. The STEREO B panels are 90° turned left from their original appearance for an easier comparison with ENLIL predictions.

Comparing the three grey panels in Figure 3b with the corresponding panels in Figure 3a, one can notice that ENLIL describes the current global situation in the solar wind rather approximately. The HI observes more ICMEs than predicted by ENLIL, the four instead of two as seen in the upper panel. The number of SIRs is five, and one can find from the original Heliowheather movies from HI STEREO B that the leading ICME actively interacts with one of the SIRs at the very early stage of its propagation. As a result, its front becomes w-shaped. Since the ICMEs interact with each other and several SIRs,

the front of the first ICMEs becomes depleted and the fronts of all ICMEs are far from radial. The SIRs tend to merge by the Martian orbit, according to the HI observations, which is not the case in the ENLIL simulations.

---

[10] http://www.helioweather.net/archive/2012/03/stb1dej.html



It is also obvious that the sequence of events realizes faster in reality. The two ICMEs caught by ENLIL actually have wider fronts and move faster than predicted. According to the bottom panel of Figure 3b, soon after the passage of the predicted "ICME flank-SIR" conglomerate, the Earth faces the other, unpredicted strong front of two merged ICMEs flanks on March 08, 2012. The time of the detection of the disturbances at the Earth's location perfectly corresponds to distant STEREO observations shown in the bottom panel of Figure 3b and poorly correspond to ENLIL predictions, as one can easily find at https://omniweb.gsfc.nasa.gov/form/sc_merge_min1.html (not shown). This result is generally in agreement with studies in which ENLIL predictions are compared with observations (e.g., Jian et al., 2011).

Therefore, although global ENLIL modeling is a tremendous step forward in understanding of the 3-D heliosphere, the real picture of large-scale processes occurring in the solar wind is too complex to be predicted in full; and more STEREO-type missions are required to control space weather and understand in situ observations. Currently, there is only one spacecraft stably providing real time remote-sensing information about the solar wind from HIs, namely STEREO A[11,12], and there are reconstructions from the ground-based interplanetary scintillation (IPS) data provided by ISEE, Nagoya University, Japan[13]. Therefore, until several missions carrying HIs ensure covering of the entire sphere around the Sun with white light observations, the development of ENLIL-like models and their synergy with available methods of heliospheric tomography are very important.

Current coronal magnetic field models (both numerical and analytical) use as boundary conditions photospheric field measurements (line-of-sight component averaged over a solar rotation, and/or actual vector magnetic fields). These measurements are then extrapolated to estimate/infer the 3D open/closed coronal and solar wind magnetic structure and the presence of high-speed streams. Refinements of these observational and modeling techniques are essential for improving space weather forecasting.

The internal structure of CIRs/SIRs as well as ICMEs is also a major open research question with significant space weather consequences. Recent studies confirm prior suggestions that CIRs/SIRs as well as ICMEs frequently embed strong current sheets and plasmoids/blobs/flux ropes (in 3D) or magnetic islands (in 2D consideration) of various origins, both local and solar (Khabarova et al., 2016; Khabarova and Zank, 2017; Sanchez-Diaz et al., 2017; Malandraki et al., 2019). In section 2.1.3 we will discuss how these local structures enhance the ability of ICMEs and CIRs to accelerate particles locally.

In this section, it is important to note that since some of these structures are created by magnetic reconnection in the solar wind, they are observed in front of the high-speed streams/flows in magnetic cavities formed by the HCS and approaching ICMEs and CIRs/SIRs (Khabarova et al., 2016; Adhikari et al., 2019). Therefore, small-scale magnetic islands (SMIs) of ~0.001-0.05 AU length may represent potentially geo-effective structures, being an underestimated source of Ultra Low Frequency (ULF) magnetospheric waves that occur at the magnetopause due to the solar wind-magnetosphere interaction. Khabarova et al. (2016) performed the Wavelet-analysis of the IMF and plasma parameters and noticed that crossings of SMI-filled regions are observed as quasi-regular variations in the IMF and the solar wind density in the ULF range, which at least partially explains the prior findings of the increased level of ULF-variations in the solar wind before geomagnetic storms.

SMIs can be geoeffective for the following reasons (see Shi et al, 2014; Liu et al. 2016 and references therein): (i) ULF pulses create or significantly modulate fluxes of so-called "killer" electrons in the Van Allen belts with energies up to a few MeV; (ii) ULF pulses of the solar wind destabilize the magnetosphere as a turbulent IMF drives auroral activity more strongly than the laminar solar wind; (iii) ULF pulses generate lower-latitude geomagnetic field variations in the ULF range and long-lived plasma vortices in the nightside magnetospheric plasma sheet, being associated with various secondary effects; and (iv) SMIs represent a source of ULF solar wind pressure variations known for their geoeffectiveness. The

---

[11] https://secchi.nrl.navy.mil/secchi_flight/images
[12] https://helioviewer.org
[13] https://ips.ucsd.edu/



presence of ULF variations in the IMF and the solar wind plasma in magnetic cavities formed by the HCS from one side and an ICME or a CIR/SIR from the other can be used for prognostic aims. For an observer at the Earth's position, such ULF variations and Atypical Energetic Particle Events (AEPEs) occur before the onset of a geomagnetic storm produced by geoeffective ICMEs or CIRs/SIRs, which may improve the accuracy of mid-term prognoses of geomagnetic storms.

### 2.1.3 Predictability of interplanetary shocks and energetic particle flux enhancements

Solar energetic particles (SEPs) with energies overlapping with cosmic ray energies present a major space weather hazard. Predictability of the occurrence of solar radiation storms, characterized by severe enhancements of the solar energetic particle flux, strongly depends on the predictability of flares and properties of CMEs (see above). In addition to their critical role in local particle acceleration, interplanetary shocks (ISs) are highly geoeffective in terms of their interaction with the terrestrial magnetosphere. At the Earth orbit most ISs are forward shocks driven by CMEs. ISs are also formed at leading/trailing edges of CIRs/SIRs, but typically far from the Earth, at 2-3 AU. Consequent differences in propagation and inclination of shock fronts to the interplanetary magnetic field direction determine peculiarities of their geoeffectiveness and the efficiency of particle acceleration, which is still investigated insufficiently. Theoretical studies of the fundamental properties of shock waves in the solar wind plasma are needed. Owing to the strong nonlinearity of processes of particle acceleration and IS-magnetosphere interaction, IS geoeffectiveness is influenced by properties of the solar wind through which an IS propagates.

According to coronographic and EUV imaging observations, CME-driven shocks can be formed at the earlier stage of CME development in the corona (e.g., Zucca et al., 2018), but they can also form at larger distances, depending on the acceleration profile of CMEs (Gopalswamy et al. 2015). After formation, ISs propagate further in the solar wind and are easily distinguishable in situ because of typical jumps in the plasma and IMF parameters at the shock front (e.g., Richardson, 2011). The following catalogues of ISs observed by different spacecraft in the solar wind can be recommended for practical aims: https://www.cfa.harvard.edu/shocks/wi_data/ , and http://ipshocks.fi/database.

Using the catalogues, one can find that ICME-driven fast forward ISs dominate over other types of shocks at 1AU except for the solar minimum phase of the solar cycle, as shown by Kilpua et al. (2015a). The cycle dependence appears because ISs are also formed at leading/trailing edges of CIRs/SIRs that prevail during solar minimum. As noted above, the CIR/SIR-driven IS formation typically takes place far from the Earth, at 2-3 AU. Differences in origination, propagation and inclination of shock fronts to the interplanetary magnetic field direction determine peculiarities of IS geoeffectiveness and the efficiency of particle acceleration, which is still investigated insufficiently.

Interplanetary models, the output of which suggest a prediction of the IS arrival to the Earth's magnetosphere, usually use the location, the duration, the class of the solar event, the total energy of the explosion, and the CME initial speed as input parameters (e.g., Smith et al., 2009), and the rest of parameters used may vary. Peculiarities of the IS-magnetosphere interaction and geoeffectiveness of ISs followed by ICMEs/SIRs significantly depend not only on the shock obliqueness or the impact angle but also on properties of the solar wind through which a particular IS propagates (e.g. Smith et al., 2009; Núñez et al., 2016). These properties include the ambient solar wind speed, characteristics of turbulence and the IS interaction with various quasi-stable solar wind structures, i.e., other ISs, streams and the HCS.

ISs associated with the X-class events are the most predictable (Smith et al., 2009), which again shifts the focus of forecasts towards extreme or at least strong events causing intense geomagnetic storms. Meanwhile, predictability of mild and medium-intensity events is still a weak point. The other point is that IS predictions suggest searching for changes in active regions, not CHs. As a result, overwhelming majority of IS prognoses are able to predict the arrivals of CME-driven shocks only.



Meanwhile, there is a prognostic technique that can take CIR/SIR-driven ISs into account. It is based on
observations of energetic particles that may have different origins (e.g., Vandegriff et al., 2005). Indeed, most of strong
ICME-driven shocks are preceded by the arrival of SEPs of energies overlapping with cosmic ray energies (from keV to
GeV, see Mewaldt et al., 2012, and Gopalswamy et al., 2012). SEPs represent a major space weather hazard, and therefore
are studied quite extensively (e.g., Reames, 2017). (e.g., Reames, 2017). Predictability of the occurrence of solar radiation
storms, characterized by severe enhancements of the solar energetic particle flux, strongly depends on the predictability of
flares and properties of CMEs (see above). At the same time, observations of steadily growing energetic particle flux with
energies above several MeV/nucleon at 1 AU always allow anticipating the arrival of an ICME preceded by an IS.

Even in the case of energetic particle flux enhancements observed only in lower energy channels (from keV to
MeV), one may suggest that the energetic particles are accelerated locally by an IS of either ICME or CIR/SIR origin.
Energetic particles with energies up to tens MeV stream from CIR/SIR-driven ISs from further heliocentric distances back to
the Earth position. This makes CIRs/SIRs as potentially hazardous as ICMEs not only because of their geoeffectiveness in
terms of their ability to trigger geomagnetic storms (e.g., Chi et al., 2018) but also because of the long-lasting periods of
energetic particle flux enhancements associated with them, which are especially important to predict in solar minimum (e.g.,
Posner et al., 1999).

Note that there is a way to distinguish between processes of particle acceleration occurring locally and distantly
(e.g., Khabarova and Zank, 2017). Spectrograms of the time arrival of 10–70 AMU ions, also known as ion speed dispersion
plots, show 1/ion speed vs time as observed in situ by a spacecraft (4-Day Ion 1/Velocity Spectrograms from the Ultra Low
Energy Isotope Spectrometer of the Advanced Composition Spectrometer (ACE/ULEIS) are available at
http://www.srl.caltech.edu/ACE/ASC/DATA/level3/summaries.html). On each spectrogram, there is the diagonal black line
showing a pattern of the velocity dispersion typical for free propagation of energetic particles from the Sun to the spacecraft
along the magnetic field line of a length of 1.2 AU. The inclination occurs because particles of larger energies accelerated at
the Sun propagate faster than particles of lower energies. At the same time, particles accelerated locally, for instance, at ISs
show vertical patterns in the spectrograms. The method is potentially useful for the IS arrival prediction but still not included
in predictive schemes.

Local particle energization in the solar wind has been attributed to ISs for a long time. Energetic particles can be
accelerated at ISs due to the diffusive acceleration (DSA) mechanism (e.g., Zank et al., 2000; Reames, 2017). Meanwhile, as
noted above, ISs are also associated with numerous dynamical effects developing both at the IS itself and far downstream
(see 2.1.2). It has been shown that dynamics of turbulence-borne structures, namely, current sheets and plasmoids/flux ropes
or SMIs in sheath regions ensure local particle acceleration that leads to the significant amplification in the energetic ion flux
(e.g., Zank et al., 2015).

Furthermore, both observational and theoretical studies suggest the occurrence of particle acceleration in regions
not related to ISs but filled with current sheets and SMIs or larger-scale fragmented magnetic clouds (Khabarova et al. 2015,
2016, 2020; Adhikari et al., 2019; Le Roux et al., 2019). Such acceleration is typically observed within magnetic cavities
formed by strong current sheets at edges of different streams/flow and/or the HCS. Energetic ion flux enhancements and
associated effects of local particle acceleration seen in pitch-angle distributions of suprathermal electrons are often
associated with the HCS and the heliospheric plasma sheet surrounding the HCS as well as near similarly strong current
sheets (Zharkova and Khabarova, 2012, 2015; Khabarova et al. 2015, 2020). Note that the occurrence of the HCS also
impacts the propagation of SEPs in the heliosphere at global scales, which should be taken into account (e.g., Battarbee et
al., 2018).

The energy range in which local effects associated with current sheets and SMIs can be expected is from tens keV
to several MeV. However, the upper threshold may be larger since it strongly depends not only on the typical size of flux
ropes but also on the energy of so-called seed particles pre-existing in the system. Combined cases of SEPs and DSA-





particles as seed particles re-accelerated locally in SMI- and current sheet-regions may be observed with energies reaching

tens MeV (Zank et al. 2015; Khabarova et al. 2016; Khabarova and Zank, 2017; Malandraki et al. 2019). Meanwhile, the

effects are often spatially separated, i.e., associated with the same stream but effective in its different parts (see Figure 4 for

the case of consequent particle acceleration by an IS and magnetic islands).

Summarizing, forecasts of SEP events should be enhanced by forecasts of energetic particle enhancements

determined by local effects. Local particle acceleration in the solar wind is a newly-found and practically unexplored

phenomenon, which definitely requires attention of the solar-terrestrial community. Both theoretical and observational

studies of the fundamental properties of shock waves and associated effects are needed. The further work in this direction

requires not only the improvement of the existing models, but also creation of new robust models that would take into

account a variable response of the magnetosphere on weak and medium intensity events that include not only CME but also

SIR/CIR associated impacts.

### 2.2 Solar-wind magnetosphere coupling, internal magnetospheric dynamics, and the predictability of substorms and
geomagnetic storms

Solar wind – magnetosphere coupling and internal magnetospheric dynamics play complex and crucial roles in geospace

weather. Accurate and reliable predictions of geospace weather require the understanding of all key aspects of the complex

interplay of external and internal regulating factors operating over timescales ranging from minutes to days. Radiation belts,

in particular, can experience drastic changes in timescales as short as minutes, while as mentioned above, a substorm cycle

lasts a few hours and a geomagnetic storm several days.

With regard to the major geospace weather disturbance, i.e. the geomagnetic storm, its drivers arrive at Earth's orbit

in characteristic sequences lasting from tens of hours to days. The most important drivers of geomagnetic storms are ICMEs

and high-speed streams (HSSs) with associated stream or corotating interaction regions (SIRs/CIRs), due to their prominent

IMF southward component and high solar wind speed. Interplanetary shocks associated with ICMEs have profound effects

on the level of geomagnetic activity (Zong et al., 2009; Yue et al., 2010; Yue and Zong, 2011).

The ambient interplanetary magnetic field will be compressed by a perpendicular shock more strongly than by a

parallel shock. Thus, a perpendicular interplanetary shock can produce more intense geomagnetic activity than a parallel

shock under the same IMF pre-condition. Furthermore, with a southward IMF pre-condition substorm onsets may be more

likely to follow the interplanetary shock arrival, while with a northward IMF pre-condition, only typical compression effects

to the magnetosphere are observed. Together with a southward IMF pre-condition, interplanetary shocks and driven ICMEs

can intensify geomagnetic storms significantly. Studies show that interplanetary shocks can intensify southward IMF Bz pre-

condition by a factor of 3 to 6. This effect would enhance geomagnetic storms greatly (Figure 5; Yue and Zong, 2011).

Moreover, strong interplanetary shocks lead to rapid and pronounced enhancements of relativistic electrons within a

few minutes. An outstanding example was the shock that triggered the sudden commencement of the 24 March 1991 storm

and compressed the magnetopause inside the geosynchronous orbit. The shock impact resulted in the rapid formation of a

new radiation belt at L≈2.5, with a peak in the electron energy spectrum at 15 MeV (Blake et al., 1992; Li et al., 1993),

observed by the Combined Release and Radiation Effects Satellite (CRRES). A similar event was observed by the Van Allen

Probes in March 2015 (Kanekal et al., 2016). A statistical study of Van Allen Probes observations showed that about 25% of

interplanetary shocks impacting the magnetosphere are associated with prompt electron energization (Schiller et al., 2016).

While general interplanetary constraints for causing significant geomagnetic storms and relativistic electron

enhancements in the outer Van Allen belt are relatively well understood, there a several open questions related to the details

of solar wind-magnetosphere coupling, such as:





- How do various solar wind conditions (e.g., IMF components, speed, density, level of turbulence) and different large-scale drivers control the coupling efficiency and the energy/mass transfer from the solar wind to the magnetosphere?


- How do solar wind conditions control the occurrence frequency and location of different magnetospheric plasma waves?

With regard to internal magnetospheric dynamics, some of the most pertinent open issues are:


- How do electromagnetic waves of various modes in the inner magnetosphere (e.g., ULF, chorus, and electromagnetic ion cyclotron (EMIC) waves) influence the acceleration, transport and losses of radiation belt electrons?
- How do both external and internal processes drive and regulate such waves and how do they determine which mechanisms dominate energetic particle dynamics?


- How do other plasma populations in the inner magnetosphere, such as the plasmasphere and ionosphere (including ion outflow) influence and contribute to energetic particle dynamics?

As these are key issues in the predictability of the geospace radiation environment, studies to address them using coordinated space-borne and ground-based instrumentation along with models are of essential importance.

Magnetospheric substorms and geomagnetic storms are the most important collective complex phenomena in


geospace, dissipating the energy transferred by the solar wind to the magnetosphere and upper atmosphere. While a substorm cycle lasts approximately 2-3 hours, a storm may last from few hours to even weeks. Substorms with significant space weather effects can also occur without magnetic storms, while the storm-substorm relation is still under debate (cf., Daglis et al., 2003; Daglis and Kamide, 2003; Runge et al., 2018).

The importance of storms and substorms for space weather relates to several aspects:


- They generate waves (through substorm-injected electrons and ring current ion anisotropies), that can accelerate electrons to relativistic energies; such electrons are the causes of internal charging of satellites and associated malfunctions (e.g., Hilgers et al., 2007; Reeves and Daglis, 2016; Daglis et al., 2019).
- They are responsible for geomagnetically induced currents (GICs), which are a serious threat for power grids (e.g., Pirjola, 2007).


- They supply energetic electrons to the inner magnetosphere, which form the seed population for relativistic electrons in the outer Van Allen belt. Seed electrons pose themselves a threat to satellites through surface charging (e.g., Garrett and Whittlesey, 2011; Sarno-Smith et al., 2016; Daglis et al., 2019).

There have been several prediction models for Dst (i.e., the intensity of the storm-time ring current) and AL (i.e., the intensity of the substorm auroral electrojet current). Due to the importance of both phenomena, the scientific community


should continue the effort of improving such prediction models. Understanding the substorm triggering mechanism, in particular, is a pertinent science research topic and should be taken into account in any prediction model. Key model items include the timing of substorm onset, and the intensity, spatial location and extent of the substorm.

### 3 Space weather and the Earth's atmosphere

The space weather of the middle and upper atmosphere (including ionosphere) is characterized by variability occurring on


timescales of minutes to weeks. This short-term variability is governed by several processes which partly originate at lower altitudes (e.g., planetary waves, atmospheric tides, and gravity waves; Oberheide et al., 2015) but also come from outside the Earth's system (e.g., solar particles and radiation; Lei et al., 2008). Anthropogenic sources can also drive changes at large and small scales in the middle and upper atmosphere (Emmert et al., 2012; Yue et al., 2015; Lin et al., 2017). The near-Earth



space weather is of critical societal importance due to, its influence on communication and navigation operations (Kelly et
al., 2014; Frissell et al., 2014), as well as forming the dominant orbital perturbation and re-entry environment for spacecraft
in Low Earth orbit (LEO) (Leonard et al., 2012; He et al., 2020).

**3.1 Response of the thermosphere and ionosphere to various forcings from above and from below**

The thermosphere and ionosphere are driven by both the upward coupling from waves originating in the lower atmosphere
and downward coupling from solar and magnetospheric forcing. Understanding the response to these forcings is critical for
specification and prediction of the thermosphere and ionosphere, and their impact on communication, navigation, and
spaceflight operations. Recent ground and space-based observations, combined with the development of whole atmosphere
models have led to increased understanding of how the thermosphere and ionosphere respond to forcing from above and
below.

A focus of this research area is to better understand the internal variability and to improve the predictability of a
variety of phenomena. At high latitudes, particle precipitation, electric field penetration, Joule heating and Lorentz force can
introduce the signatures of both the longer scale solar variability and of the shorter time scales of recurrent geomagnetic
storms (Lei et al, 2008; Deng et al., 2008). At mid to low latitudes, plasma irregularities and travelling
ionosphere/atmospheric disturbances can result in the diffraction of trans-ionospheric radio signals used for satellite
navigation and communications, resulting in the scintillation and fading of the received signals on the ground (Kelly et al.,
2014; Tsunoda et al., 2015). The equatorial electrojet, equatorial ionization and temperature anomalies dominate the
structures of the ionosphere in this region (England, 2012; Liu et al., 2017), while the equatorial thermosphere anomaly is
manifested in the thermosphere neutral density and temperatures (Lei et al., 2012; Liu et al., 2017). These features, as well as
the global ionospheric ($Sq$) currents are sensitive to the effects of vertically propagating atmospheric tides (Yamazaki et al.,
2016). Stratospheric sudden warmings (SSWs) have also been found to alter the local time variation of the ionosphere
(Goncharenko et al., 2010; Lin et al., 2012). This has been attributed to variations in upward propagating atmospheric tides
in the ionospheric E region wind dynamo induced by altered propagation conditions in the stratosphere and mesosphere (Liu
and Richmond, 2013), and to amplification of the lunar gravitational tides (Forbes and Zhang, 2012). The thermosphere is
also known to exhibit considerable variability during SSWs due to tidal induced changes in the thermospheric mean
circulation (Pedatella et al., 2016), and changes in gravity wave drag (Yiğit et al., 2014). Specific questions related to this
focus area include:

1.  What is the response of the thermosphere and ionosphere to magnetospheric forcing?
2.  What is the influence of the lower atmosphere on ionosphere and thermosphere dynamics?
3.  What are the factors controlling the occurrence of equatorial plasma irregularities and what are their relative
    importance?
4.  How and in what ways do solar flares modulate the terrestrial atmosphere?
5.  What are the technological consequences (e.g., GNSS positioning, radio wave propagation, satellite drag) of
    ionospheric and thermospheric variability?

The community should continue to work on advancements of the aforementioned science issues in order to enhance
understanding of the predictability of vertical and horizontal coupling produced by factors from above and below the
ionosphere/thermosphere region.

**3.2 Impact of atmospheric waves and composition changes on the middle and upper atmosphere**

The results from the past decade have shown that wave sources originating in the lower atmosphere, such as tides, planetary
and gravity waves can have a significant and persistent effect on the variability and structure of the middle and upper





atmosphere (Oberheide et al., 2015). These sources can be generated by convection and jet streams in the lower atmosphere, SSWs in the middle atmosphere, as well as wave breaking and mixing in the upper atmosphere. These waves span a large spatial and temporal spectrum, ranging from small-scale gravity and acoustic waves with durations of minutes, to the global-scale tides and planetary waves which vary on daily, seasonal, and interannual time scales. As they propagate upwards, atmospheric waves can have a considerable impact on the middle and upper atmosphere. This can be through wave

dissipation, in which breaking waves and tides impart momentum forcing upon the background winds (Chang et al., 2011), while also enhancing the eddy mixing of the normally diffusively separated thermosphere and ionosphere above the turbopause (Qian and Solomon, 2012; Yamazaki and Richmond, 2013; Yue and Wang, 2014; Chang et al., 2014). The vertically propagating waves themselves also represent an important source of variability in the middle and upper atmosphere, with some very long vertical wavelength tides and waves being capable of propagating well into the upper

thermosphere (Oberheide et al., 2015; Gasperini et al, 2015). Anthropogenic sources of waves and composition change have also been shown to be sources of middle and upper atmospheric variability. Elevated carbon dioxide levels diffusing upwards from the lower atmosphere have been shown to manifest in both the mesosphere and thermosphere (Emmert et al., 2012; Yue et al., 2015). Smaller scale travelling ionospheric disturbances have also been found to be generated by the wake waves produced by satellite launch vehicles (Lin et al., 2017).

Of critical importance is understanding how the wave spectrum evolves with altitude, and its consequent impacts on the middle and upper atmosphere. Among the key questions to be considered are:

1. How do we quantify the effects of gravity waves, planetary waves, and tides (and their interactions) on the dynamics and chemistry of the middle and upper atmosphere?
2. What is the extent that SSWs couple the whole atmosphere, including effects on dynamics, composition, and chemistry?
3. Can gravity waves be better defined in terms of the mesoscale GW spectrum, amplitude, and vertical penetration into the thermosphere?
4. What is the predictability of atmospheric waves and their effects on the middle and upper atmosphere?
5. How do the various waves contribute to the global dynamics of the thermosphere and ionosphere?

Long-term changes in atmospheric composition (e.g., $CO_2$, $O_3$) may also have consequences for the short-term variability of the middle and upper atmosphere. This could either occur directly through changes in the wave sources and forcing, or indirectly by changes in the mean flow which impact the wave propagation. The thermosphere is additionally radiatively cooled following geomagnetic storms by $CO_2$. An increase in baseline $CO_2$ levels thus has the potential to lead to a different response of the thermosphere to geomagnetic disturbances (Emmert et al., 2012; Yue et al., 2015; Lin et al.,

2017). Questions that may be addressed related to the role of changing composition on the middle and upper atmosphere variability include: 1. How does long-term changes in composition impact the wave spectrum in the middle and upper atmosphere? 2. To what extent do changes in $CO_2$ influence the radiative cooling of the upper atmosphere?

**3.3 Magnitude and spectral characteristics of solar and magnetospheric forcing**

Earth's atmosphere and ionosphere are significantly affected by the increased energy deposition that occurs during solar and magnetospheric driven disturbances. Solar flares cause significant changes on the ionosphere and thermosphere. Geomagnetic disturbances, including storms and substorms, provide energy input into the high-latitude atmosphere which eventually propagate to lower latitudes. Various magnetospheric processes, such as plasma waves and pitch angle scattering, cause precipitation of the energetic particles from the magnetosphere to the ionosphere and neutral atmosphere. These

different sources of energy inputs into the ionosphere and atmosphere are complex, and modelling/observation efforts are necessary to improve specification of the energy inputs. Predictive skill of the mesosphere, thermosphere, and ionosphere



depends in part upon accurate specification and prediction of the different solar and magnetospheric forcing. Critical areas to be addressed include:

1. What is the magnitude, spectra, and location of particle precipitation from the magnetosphere to the ionosphere and atmosphere?
2. What are the global electric currents and electric fields imposed from the magnetosphere to the ionosphere?
3. How to predict and quantify the likelihood of occurrence of solar flares and their spectral characteristics?
4. What is the uncertainty in specifications of solar and magnetospheric forcing?

**3.4 Chemical and dynamical response of the middle atmosphere to solar and magnetospheric forcing**

Energetic particle precipitation (EPP) from solar eruptions, galactic cosmic rays, and Earth's magnetosphere penetrate into the atmosphere. EPP deposit energy and trigger local ionization which perturbs the chemical and thermal structure of the middle atmosphere at high latitudes. Notably, EPP leads to production of nitrogen oxides (NOx = {NO + NO$_2$}) and hydrogen oxides (HOx = {H + OH + HO$_2$}) that can strongly contribute to ozone depletion in the mesosphere and stratosphere (Seppälä et al., 2014). Ozone changes could then affect the radiative balance which in turn would modulate the
atmospheric dynamical state of the middle atmosphere. These changes in the dynamics of the middle atmosphere will affect the propagation of waves into the upper atmosphere, thus providing another pathway in which EPP can affect the upper atmosphere and ionosphere. Though strong EPP events are themselves relatively short-term processes, EPPs may introduce longer term variability due to the differences in their frequency over the solar cycle (the most extreme events often occur in a few years after solar maximum). For this reason, EPPs have been recognized as one part of the solar-climate connection by
the climate community, and have been recently included in the CMIP6 (6th Phase of the Coupled Model Intercomparison Project) recommended solar forcing (Matthes et al., 2017), and are important for Pillar 3 of PRESTO.

The source region of EPP generated NO$_x$ and HO$_x$ is the mesosphere and lower thermosphere (MLT, 50-150 km), an altitude region not included in most climate models. A better representation of the MLT is required in order to improve our understanding of the EPP effect on the middle atmosphere and climate. Although significant progress has been made in
recent years, there remain discrepancies of an order of magnitude (or more) between modeled and observed NO$_x$ (Andersson et al., 2018). Potential reasons of the model-observation discrepancy include inaccurate magnetospheric inputs, ionization rates, and underrepresentation of the downward transport. Evaluating the reasons for these discrepancies is critical for improving the representation of EPP in climate models. The complex chain of coupling processes that have an end result of dynamical changes in the lower-middle atmosphere, as illustrated in Figure 1, are also not well understood (e.g., Seppälä et
al., 2013).

**4 Solar activity and its influence on climate**

**4.1 Solar activity: Understanding the past and predicting the future**

The next five years in the run-up to the maximum phase of Solar Cycle 25 provide an excellent opportunity for understanding solar cycle predictability and assessing data-driven (MHD dynamo models of the solar cycle. Decadal
timescale activity is typically parametrized in variations of the sunspot number or surface magnetic flux that can be simulated by data driven solar dynamo models. Surface flux emergence and its evolution driven by flux transport processes govern the Sun's polar field reversal, distribution of open and closed magnetic field lines and the large-scale structuring of the corona. These models are now capable of separately predicting the Northern and Southern hemisphere activities which may be used for assessing asymmetry related impacts on the heliosphere. Space weather and climate drivers, such as the
frequency of coronal mass ejections (CMEs) and flares, spectral and total irradiance variations, open flux variations and





cosmic ray fluxes expected over decadal timescale may be derived from these dynamo and surface flux transport model-based predictions.

Quasi-periodic bursts in solar activity, manifest in sub-annual to annual-scale. Short-term fluctuations are also apparent in the sunspot time series which may have important space weather consequences. Understanding and predicting

these quasi-periodic variations may therefore benefit short-term space weather and long-term space climate assessment. A dynamical memory on the order of solar rotation timescale exists in the large-scale coronal structure which may be used for predicting the evolution of global coronal and heliospheric field up to a month ahead. This may allow similar time windows for predicting the structure and strength of the solar wind, interplanetary (open) magnetic flux and cosmic ray fluxes.

On shorter timescale of days, both active region properties and MHD simulations are currently generating

likelihood predictions of flares, CMEs and solar wind conditions, which are being used by operational space weather agencies, e.g., NOAA Space Weather Prediction Center. These necessitate continuous measurements of vector magnetic fields of solar active regions and exploring which near-Sun properties determine eruptive potential. Machine learning techniques are beginning to be applied to these data-based approaches. Computational approaches include data-driven coronal field modelling techniques that are becoming more complex and sophisticated with increasing computing power.

Uncertainties remain in terms of a) the underlying assumptions in dynamo models and differing predictions (e.g., solar cycle 24), b) prediction of the timing and properties of solar eruptions and c) seamlessly bridging different timescales. Solar cycle predictability beyond a decade also remains a major open question and some studies indicate this is not possible. Will we have a solar cycle 25 or will there be an imminent slide to a Maunder Minimum like phase? A critical comparative assessment of theoretical-computational models of solar activity, testing their underlying assumptions, and confronting them

with past data may lead to transformative progress in understanding and predicting solar activity in the next decade. Such advances would enable accurate, physics-based inputs from the Sun to global climate models.

Assessing how solar activity models perform requires their testing with historical datasets. Reconstruction of past solar activity and long-term climate variations (across centuries) also opens up the possibility of separating natural and anthropogenic causes of climate change. In the industrial and post-industrial era, anthropogenic forcing clearly dominates

over natural climate drivers and thus going back to the pre-industrial era to establish the role of natural drivers is crucial. However, large uncertainties remain. There are information gaps, e.g., for past solar spectral irradiance variations over millennial timescales and the floor of activity during the Maunder Minimum.

Reconstructing long-term solar impacts on Earth's climate is also difficult. Distinguishing between t solar driven regional as opposed to global climate impacts, in sparse historical records is challenging, but necessary if we are to

understand solar driven impacts on large scale atmospheric and ocean circulations. These questions need to be addressed to understand and assess the system-wide impact of solar variability. The emphasis should be on deciphering the physical pathways of Sun-Climate relationships, e.g., what physics of atmospheric systems is impacted by solar variability, rather than focusing simply on the global temperature, which is a net outcome of diverse factors.

## 4.2 Sub-seasonal to decadal variability of the terrestrial system

A grand challenge in environmental prediction is to bridge the gap between the weather and climate timescales. The sub-seasonal to seasonal (S2S) to decadal timescales are of particular interest. These are the timescales that are considered most relevant by policy makers and drive decisions in terms of, e.g., infrastructure investments or land use. Forecast systems can already predict weather out to several weeks with reasonable accuracy and variations on centennial scales are well represented in climate models. It seems reasonable to assume that some progress can be made in the intermediate timescales

if we can simultaneously improve the forcing of the Earth system as well as the understanding of its response. Better prediction of the solar and geomagnetic forcing, with their inherent 11-year variations, and improved characterization of the



atmosphere-ocean response to that forcing could be one way to make progress, and one of the objectives of this program. For space weather (see section 1) the timescales are much shorter. Further, there are good reasons to believe improvements in geospace prediction (especially under quiet solar conditions) could come from better characterization of the forcing from below. For example, stratospheric vortex variations (such as SSWs) have timescales on the order of weeks and it has been shown that they affect the ionosphere, the troposphere and surface weather and climate through their interactions with upward propagating waves. This driving should be integrated into space weather forecast systems. Further progress can also be made via data assimilation to create improved initial states for forecast systems. It is critical to understand where the data and knowledge gaps are and try to address them. Expected societal benefit could be used to prioritize research efforts.

### 4.3 Solar activity and its influence on the climate of the Earth System

#### 4.3.1 Solar influence on climate

Better prediction of the solar and geomagnetic forcing, with their inherent 11-year variations, and improved Analysis of the Earth's climate history suggests that solar activity variations contribute to climate variability on decadal-to-centennial timescales. However, the magnitude of this influence and the key responsible mechanisms remain to be quantified. Several pathways are proposed to explain the influence of solar variations on regional climate. Among them, the "bottom-up" pathway refers to climate perturbations induced by fluctuations of the solar energy input which directly reaches the Earth's surface. Alternatively, the "top-down" pathway invokes solar-induced changes in the middle atmosphere (through solar UV irradiance changes or energetic particle precipitation) that in turn affect regional climate through stratosphere-troposphere couplings. Causal connections in both these pathways need to be explored and determined; accurate identification and attribution of their impact on climate remains elusive. One of the main challenges is to determine how low-frequency variations of solar activity influence, and/or interact with, the coupled ocean-atmosphere system which intrinsically varies at decadal-to-centennial timescales.

Adequate representation of the complexities of the coupled atmosphere-ocean-sea ice system in numerical models is required to better understand and quantify the solar influence on climate. In addition, these climate models should ideally resolve the entire middle atmosphere and calculate ozone chemistry interactively as both are key components of the "top-down" pathway. Finally, model experiments need to be sufficiently long and repeated to ensure the robustness of results. To date, meeting these requirements has been nearly impossible. However, the increase of computing power, novel data mining and machine learning techniques, and improvements in climate models offer new opportunities to numerically explore the Sun-Climate relationship and make future projections. Transformative progress in these fronts may be achieved by coordinating efforts and bridging climate modelling, paleoclimate reconstructions, space weather and solar physics communities.

#### 4.3.2 The impact of increasing radiatively active gases on the middle and upper atmospheric response to solar variability

The ITM system is evolving to a fundamentally new state due to the continued buildup of carbon dioxide in the atmosphere as a result of human activity. These changes, which are already becoming apparent, will have profound effects on the structure and composition of the ITM system, and potentially, on the long-term 'habitability' (i.e., the sustainability of its use) of low-earth orbit. Increasing carbon dioxide will ultimately cool the entire ITM system (as well as the stratosphere) and will result in density decreases approaching 5-8%/decade under solar minimum conditions at satellite altitudes. In addition, $CO_2$ increases it will change the cooling rate of the thermosphere and the timescale by which the atmosphere dissipates solar



storm driving (as described above). Both these processes will introduce a long-term trend in the way the atmosphere responds to space weather forcing. Finally, the effects of a cooler thermosphere on the chemistry of NO are also important given its production during geomagnetic storms and role in dissipating storm energy. Changes in the abundance of NO in a

cooler thermosphere, its effects on storm dissipation times, and storm time thermospheric density are also crucial in the prediction of thermospheric variability.

Discerning the evolution of the ITM system is, in principle, a problem in trend detection, but one that is inherently tied to solar variability and solar-terrestrial physics, as well as to the variability of the lower atmosphere. This driving of the ITM system from "above" and "below" provides the natural variability (that is, 'noise') in the ITM system from which the

trend signals must emerge to be detected. If we were to enter an extended period of weaker solar activity (as speculated in some quarters) this would reduce the natural variability of the ITM system. In addition, long-term changes in the troposphere may alter the variability of the ITM system due to forcing from below. Understanding the effects of the natural forcing, and how they will influence the detection and prediction of long-term change in the ITM system, is a daunting problem in solar-terrestrial science.

Prediction of long-term ITM changes is more than just a pure scientific interest. As density changes at satellite altitude, lifetimes of all orbiting objects, including debris, increase significantly. With the projected launch of thousands of satellites over the next decade debris will proliferate, posing a threat to the habitability of regions of low earth orbit. Predicting the long-term temperature and density changes resulting from trends in greenhouse gases and the dependency of those trends on the solar cycle will influence international space policy for the rest of this century and beyond. It will be a

major factor in the satellite insurance and re-insurance industry.

**5 The grand challenge questions**

For each research pillar of PRESTO we have identified four grand challenge questions, which are listed below.

For pillar 1 (Sun, interplanetary space and geospace)

• Under what conditions are solar eruptions, CMEs, and SEPs produced, and which indicators of pre-CME and pre-flare activity are reliable?

• What are the required model input parameters to successfully forecast the arrival of SEPs and the geoeffectiveness of CMEs, SIRs/CIRs and the consequences of their interactions?

• How are different magnetospheric disturbances and waves (which are critical for the ring current and radiation belt

dynamics) driven by solar wind structures and variations, and/or internal magnetospheric processes?

• How can we improve the predictability of geomagnetic storms, substorms and particle radiation enhancements, allowing forecasting of their impact on both the space environment and on infrastructures on the ground and in space?

For pillar 2 (Space weather and the Earth's atmosphere)

• How does the thermosphere and ionosphere respond to various forcings from above and from below?

• How do atmospheric waves and composition changes impact the middle and upper atmosphere?

• What is the timing, magnitude and spectral characteristics of solar and magnetospheric forcing that is needed for accurate predictions of the atmospheric response?

• What is the chemical and dynamical response of the middle atmosphere to solar and magnetospheric forcing?


For pillar 3 (Solar activity and its influence on the climate of the Earth System)




- How will future solar activity vary over timescales relevant for the forcing of the Earth's climate and atmospheric dynamics?
- What is the role of coupling between atmospheric regions in the realization of the long-term solar influence on the Earth system?
- How is the atmospheric response to the variable solar forcing affected by, and interacts with, increasing greenhouse concentrations?
- How can solar activity predictions be used to improve atmospheric prediction on sub-seasonal to decadal timescales?

*Acknowledgments*. The authors acknowledge the International Space Science Institute, which provided support for two Fora, one at ISSI Beijing and one at ISSI Bern. The authors acknowledge the contributions of the following scientists, who participated in the two ISSI Fora: Amal Chandran, Seth Claudepierre, Katya Georgieva, Mamoru Ishii, Petra Koucká Knížová, Kanya Kusano, Vladimir Kuznetsov, William Liu, Franz-Josef Luebken, Shinobu Machida, Takahiro Obara, Duggirala Pallamraju, Nick Pedatella, Eugene Rozanov, Nandita Srivastava, Alphonse Sterling, Manuela Temmer, Chi Wang, Yuming Wang, and Yihua Yan. LCC was supported by grant 109-2636-M-008-004 from the Taiwan Ministry of Science and Technology. NG is supported by NASA's Living with a Star program. OVK was partially supported by RFBR grant 19-02-00957.

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





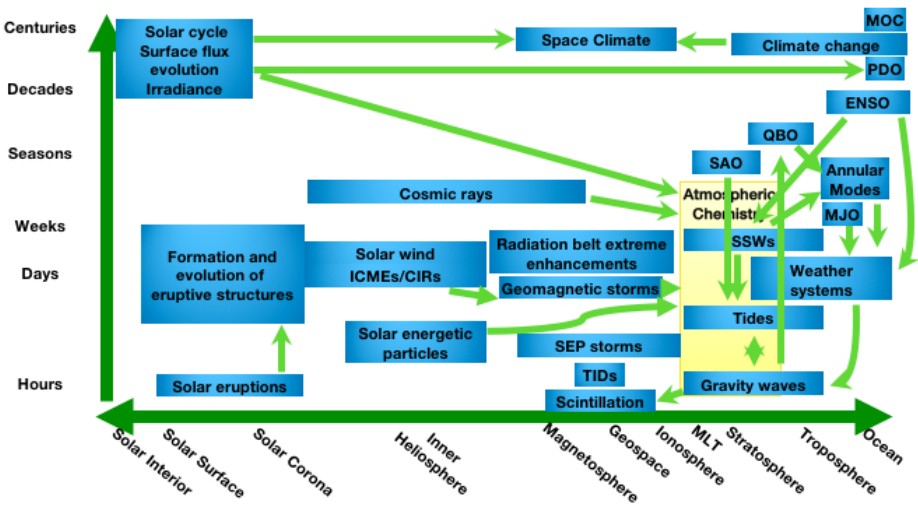

**Figure 1** An integrated view of solar-terrestrial phenomena in various spatial and temporal scales.

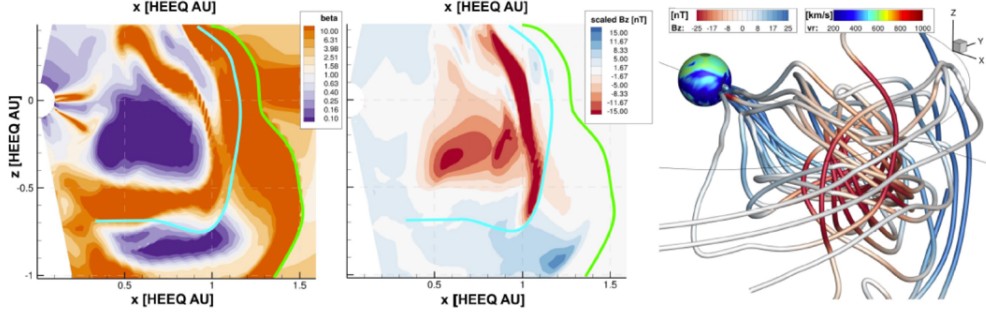

**Figure 2** EUHFORIA simulation with spheromak model for three interacting CMEs on September 7, 2017 at 18 UT. Lime
contour outlines two first CMEs that merged close to the Sun, while blue contour marks the third CME that reached the
previous CMEs further out in interplanetary space.



**Figure 3** Propagation of streams and flows and formation of magnetic cavities in the inner heliosphere in March 5-8, 2012
according to ENLIL density reconstructions based on solar synoptic charts (a) and STEREO B observations of the solar wind
in white light with the heliospheric imager (b). a) Left panel is a view in the ecliptic plane, and right panel is a longitudinal
cut at the Earth's location. SIRs resemble rotating sleeves and ICMEs are initially half-circles. The dashed white-and-black
line is the intersection of the 3-D HCS with the ecliptic plane. Red and blue colors round the circle indicate the dominant
direction of the IMF. b) Normalized density values are shown according to the scale above the upper panel, from light grey
to black. It is obvious that actual stream leading fronts have more complex shapes and propagate faster than predicted in (a).
The observed number of streams and flows is also larger. Large-scale magnetic cavities are formed by strong current sheets
at leading stream/flow fronts and the HCS.





**Figure 4** Energetic ion flux enhancements caused by different mechanisms as observed in the same stream. From top to bottom: the IMF strength, and plasma parameters (density and speed) from ACE (at L1), ion flux from ACE LEMS30 and LEMS120 (see corresponding energy channels indicated). The first ion flux enhancement is clearly associated with the IS



(blue line) as seen in lower energy channels. The role of DSA decreases with increasing energy, but, beginning with ~0.6 MeV, local particle acceleration is seen in the fragmented magnetic cloud region filled with flux ropes/plasmoids/magnetic islands and strong current sheets. Adapted from Khabarova and Zank (2017).

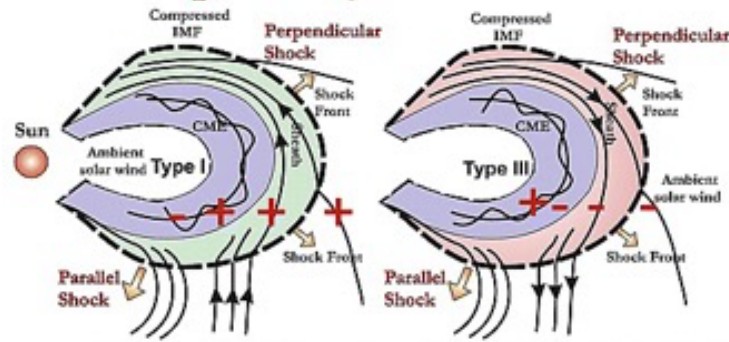

Figure 5 Schematic diagram of the Type I and Type III IP shocks and driven CMEs (Yue and Zong, 2011)
