# Peer review of "Predictability of the variable solar-terrestrial coupling"

_Annales Geophysicae, 2020_

## Referee Comment (RC1)

This review paper aims to introduce the next scientific program (NSP) of SCOSTEP. Title of the NSP is Predictability of the variable Solar-Terrestrial coupling (PRESTO). This paper is well written as it follows detailed discussions by the SCOSTEP members. This reviewer has no major comments, and recommends publication of this paper. Followings are minor comments from the reviewer, which may be considered while refining the paper before publication.

1)  The paper describes the whole program PRESTO. The style of the description is to list major questions in each research area. SCOSTEP selected this style, and the reviewer accepts. But we should be aware of its weakness. The descriptions remain qualitative, and it is hard to find quantitative discussions throughout the paper. We should have many following papers (or research projects) that propose concrete research plan or vision with quantitative targets to answer questions of PRESTO.

2)  Section "5. The grand challenge questions" may be too simple just listing the questions. It may be better to add one more sentence to show how PRESTO would be implemented in the selected period. (Additional Q: Is this PRESTO period shown in the text?)

3)  Balance between three pillows is unfortunately not very good. Descriptions of Pillow 1 may be too long compared with others. The reviewer understands that people tend to write more at the recently advanced research areas.

4)  At Line 62-67, it may be better to show like this; "Extreme events ⋯ affects modern technology ⋯ in space and on ground (e.g. spacecraft anomalies, the loss of transformers). ⋯ Strong and intense storms ⋯ have significant, although less deleterious consequences."

5)  At Line 125, "onset, onset" → "onset."

6)  Subsection 3.2 and 3.3 may be better reversed in order because the description of this paper is principally in "from Sun to Earth" manner.

7)  What is "geospace"? Is this word defined in the text? What is the difference between "space weather" and "geospacer weather"? At least in Section 2 (or in Subsection 2.2), it may be better to replace "geospacer weather" with "space weather".

8)  "ITM" (first appearance at Line 696) is not defined.

9)  At Line 710-720, ITM environmental change is discussed. Satellite insurance is listed as an example of possible changes. But design of the satellite itself or lifetime of the satellites should be affected more by the ITM change.

10) At Line 648, "between t solar" what's "t"?

11) At Line 672, "Analysis" → "analysis"

12) At Line 689, "Sun-Climate" → "Sun-climate"

13) At Line 701, "CO2 increase it will change", better remove "it".

14) At Line 704, "Changes of NO" may be better to clarify by "Increase" or "Decrease" (Sorry I do not know which is correct. Please select the correct word).

15) At Line 705, "cooler thermosphere, its effects on" → "cooler thermosphere affects"

---

## Author Response (AR1)

Dear editor,

The detailed comments of the two reviewers have helped us to improve the manuscript considerably. We have implemented most of their suggestions in the text, except for the suggestion on "adding a paragraph describing the plan of action going forward", which was not part of the SCOSTEP mandate and therefore it had not been addressed by the NSP committee. It remains to be published by the teams who now have the responsibility of the PRESTO implementation.

Best regards
Ioannis Daglis